# OpenReview forum: "Beyond Red-Teaming: CogNIP Reveals Human-Like Vulnerabilities in Safety- Aligned LLMs"
_ICML.cc/2026/Conference — Submitted to ICML 2026_

### Official Review · Reviewer_qDuz · 2026-02-25

**Soundness:** 2
**Presentation:** 3
**Significance:** 3
**Originality:** 3
**Overall Recommendation:** 4
**Confidence:** 5

**Summary:**

CogNIP is a two-stage jailbreak framework informed by cognitive science. The first stage immerses an LLM in a fictional narrative with an antagonist role, using framing effects and responsibility diffusion to elicit harmful content wrapped in story elements. The second stage asks the model to summarize that content into actionable instructions, exploiting its tendency to stay consistent with what it already generated. Across LLMs and harm categories, attack success rates jumps from about 2% with direct prompts to around 71% using the framework. The key takeaway is that cognitive manipulation can bypass alignment that holds up against straightforward attacks.

The main contribution is formalizing a connection between cognitive science and adversarial LLM testing, moving beyond ad hoc prompt engineering. The per-category analysis reveals uneven safety profiles across domains (the "Swiss cheese" pattern), and the evaluation methodology is credible, with their LLM judge validated against human experts at an intraclass correlation coefficient above 0.9.

**Compliance With Llm Reviewing Policy:**

Affirmed.

**Final Justification:**

The interdisciplinary framing is a welcome contribution to the safety literature, and the two-stage design surfaces interesting observations (e.g., partial alignment recovery at Stage 2) that single-turn setups would miss. The evaluation methodology, particularly the ICC-validated human adjudication, shows care.

However, the paper's core limitations were acknowledged but not resolved in the rebuttal. The only baseline is a direct malicious query, making it impossible to attribute the gains to the cognitive-science grounding rather than to multi-turn elaboration generally. No ablations isolate individual cognitive mechanisms, undermining the framework's explanatory power. Per-cell sample sizes (three to six trials) limit fine-grained interpretability. The theoretical framing remains stronger than the evidence warrants, though the authors have committed to softening these claims.

The rebuttal reinforced my prior assessment. The paper is technically competent and addresses a relevant problem, but the empirical gap, particularly the missing baselines and ablations, prevent it from making the case that cognitive-science-inspired design is materially better than existing multi-turn approaches. Again, this is a substantial claim to make, it requires support of equal substance. A revised version addressing these would be substantially stronger. I maintain my score of 4 (Weak Accept).

**Key Questions For Authors:**

1.	Have you evaluated CogNIP against other multi-turn jailbreak methods like Crescendo, PAP, or the cognitive overload approach you cite? The current baseline (direct malicious query) is the simplest possible comparison. If CogNIP meaningfully outperforms existing multi-turn approaches, that would substantially strengthen the soundness and significance ratings. If it performs comparably, the contribution shifts from "this works better" to "this provides a theoretical vocabulary for understanding why multi-turn attacks work," which is still valuable but should be framed differently. Or is this planned for Future Work?

2.	The supplementary spreadsheet contains summary labels rather than actual model responses for 550 of 600 trials. Are you able to provide the full generated text, even if redacted for the most operationally sensitive content? Being able to inspect actual outputs would help validate judgment labeling.

3.	Your framework bundles several psychological principles together (framing, responsibility diffusion, cognitive load, narrative transportation) in Stage 1. Have you run any ablations isolating individual mechanisms, for example narrative framing without persona assignment, or persona assignment without the high-density stylistic constraints? It would help in providing understanding around which components are doing the heavy lifting.

4.	Given footnote 2, which appropriately acknowledges that LLMs do not possess actual cognition, how do you reconcile this with the paper's title and repeated claims that the results "reveal human-like vulnerabilities"? Would you be open to reframing this as cognitive-science-inspired attack design rather than evidence of cognitive phenomena in LLMs? This is largely a framing question but it matters for how the contribution is positioned and evaluated by the community.

5.	The main paper redacts operational details in Figure 2 for safety, but AttackPrompt.pdf in the supplementary materials is fully unredacted. Is this intentional? If the paper is accepted, what is the plan for the supplementary materials? This doesn't affect the technical evaluation but it's worth clarifying given the subject matter.

**Limitations:**

Not fully, but efforts were made. The paper includes an ethical footnote on page 1 stating the work is for academic research purposes, and footnote 2 hedges appropriately on the "human-like" framing. These are reasonable but minimal, in contrast to the paper title. There is no dedicated limitations section, and the ICML impact statement requirement appears to be missing. What I would like to see is an honest limitations section acknowledging the small sample sizes and the interpretive gap between the theoretical claims and what the evidence supports with a tie in to future work.

**Strengths And Weaknesses:**

**Strengths**

Most jailbreak papers are essentially prompt engineering with post-hoc justification. Starting from established psychological mechanisms (framing effects, foot-in-the-door, cognitive load theory) and moving them into a testing pipeline is contributive to the field. We need more interdisciplinary work. Whether or not we buy the claim that LLMs truly exhibit "human-like cognitive vulnerabilities" (I do not), the framework itself is well-motivated with connection to theory.

The experimental scope is respectable. Recent frontier models, defined harm categories, and large sample trial run. The inclusion of reasoning-enhanced inference modes where available is something most papers in this space skip (though this may be changing at this conference). Figures used (such as the heatmaps) tell a richer story than aggregate numbers alone. It is made clear that no model is uniformly safe and each has distinct blind spots, this is important for the safety community.

The two-stage design is developed. Separating narrative immersion from decontextualized extraction lets the authors isolate different failure modes and makes the framework more analytically transparent than single-shot approaches. The observation that some models "wake up" at Stage 2 (Gemini on hate speech, for instance) while others maintain high compliance is an interesting finding that wouldn't surface in a single-turn setup.

Using human expert adjudication for final ASR rather than relying entirely on automated scoring, plus validating the LLM judge with ICC analysis on samples, shows rigor. The supplementary materials reinforce this: the trial-level data covers the individual trials with per-trial judgments, the HS sampling pattern matches what the paper describes, and the ICC validation data includes all response texts alongside scores from the LLM judge and human raters. The aggregate numbers appear valid.

**Weaknesses**

The sample size per cell is a concern. With six trials per model-category combination (sometimes three), individual cells in the heatmap are determined by very few observations. The difference between 66.7% and 83.3% ASR is one trial. The authors are transparent about this, but it limits how much weight we should put on fine-grained comparisons between models or categories. Some confidence intervals or significance tests would strengthen the claims considerably.

The paper's central theoretical claim, that these results demonstrate "human-like cognitive vulnerabilities," is overstated. The cognitive science concepts provide a useful design vocabulary for constructing attacks, but showing that a multi-turn narrative prompt works better than a direct malicious query doesn't actually prove the model is experiencing something analogous to the framing effect or responsibility diffusion. It could simply be that wrapping harmful requests in enough fictional context dilutes the pattern-matching that safety training relies on. The authors acknowledge this in a footnote (footnote 2) but then proceed to make strong cognitive claims throughout the paper anyway. More care in distinguishing "inspired by cognitive science" from "evidence of cognitive phenomena in LLMs" would improve the paper's credibility.

The baseline comparison feels incomplete. The only baseline is a direct malicious query, which is a covered attack type and one of the simplest (direct) possible attacks. Comparing against other multi-turn or role-play jailbreak methods (Crescendo, PAP, cognitive overload approaches they cite in related work) would give a much clearer picture of whether the cognitive science grounding actually adds something beyond what existing techniques achieve. Without this, it's hard to know if the gains come from the theoretical framework or just from the fact that any sufficiently elaborate multi-turn prompt will outperform a blunt direct attack. It's hard to tell.

The harmfulness scoring, while better validated than most, has a sampling issue. HS was not computed for every trial but rather a stratified subset, and the paper is somewhat vague about exactly which responses were scored. The 50-sample ICC validation is decent but not large. Using Gemini 3 Pro Preview as the judge while also evaluating it as a target model is a potential conflict, though the authors note that ASR is determined by human review. It would be good to more explicitly call out the delineation here to avoid any room for confusion.

The supplementary materials reveal a transparency gap around the actual model outputs. The detailed results spreadsheet contains only short summary labels for each trial ("Refused," "Provided recipe but within the plot," etc.), not the actual generated text. Only the 50 ICC samples include full responses. This means we can verify the aggregate numbers add up, but cannot independently assess whether judgment labels were applied correctly across the other trials. I firmly believe in fully transparent research, readers should be able to inspect judgement labelling.

Presentation is generally clear but the formalization in Section 3 feels heavier than it needs to be. The triplet notation and equations add a veneer of rigor without buying much analytical power since the actual attack construction is narrative and qualitative rather than optimized over a formal objective. The case analysis in Section 3.4 and Figure 2 are more illuminating than the equations. Not a major weakness, more a subjective observation.

On a related note, the paper redacts operational details in Figure 2 for safety reasons, but the supplementary materials provide every prompt fully unredacted; the inconsistency is a bit odd. Is the intent to publish this paper without the source code?

The abstract also runs a bit long relative to the suggested guideline, though this is minor.
Impact Statement required by ICML is missing, though there is a footnote on the first page that touches on ethics.

---

> ### Author Rebuttal · Authors · 2026-03-26
>
> Sincerely thank you for your review and constructive feedback!
>
> We appreciate the recognition of our work’s contributions and have carefully considered all concerns. Here are our responses to some of your critique:
>
> __Baseline comparison__
> - Response for Q1.
> - We agree that comparing against stronger jailbreak baselines (e.g., Crescendo, PAP, cognitive overload approaches) would further strengthen the evaluation.
> - Our current goal was to isolate the effect of cognitively motivated design under a controlled setting, hence we used direct malicious queries as a minimal baseline to clearly demonstrate the gain. That said, we acknowledge that this does not fully disentangle whether the improvement comes from the cognitive-science-inspired design or from the general multi-turn structure. We will include comparisons with representative multi-turn jailbreak methods in future work to better position CogNIP in the broader landscape.
> - However, our contribution is complementary: rather than proposing yet another prompt pattern, we aim to provide a principled design framework.  We will explicitly note this limitation in the revised paper and add it to Limitations section, and frame this as a key direction for future work. We thank the reviewer for pushing us to be more precise about the nature of our contribution.
>
> __Transparency of outputs and Reproducibility__
> - Response for Q2 and Q5.
> - We appreciate your emphasis on transparency. Our experiments are conducted in a black-box setting via direct interaction with deployed LLM systems, and do not rely on custom training pipelines or model-side implementations. As such, there is no standalone codebase in the conventional sense. The current supplementary materials include summary labels for all trials and full outputs for the ICC validation subset.  Due to safety concerns around releasing actionable harmful content, we limited the exposure of full responses. We agree that verifiability is important. To support reproducibility, we will explore releasing redacted versions of model outputs (with operational details removed) or providing controlled access for research purposes, to balance transparency and safety.
>
> __Ablation of components__
> - Response for Q3.
> - Thank you for this suggestion. We agree that isolating individual mechanisms (e.g., narrative framing vs. persona assignment) would provide deeper insight into which components drive the effect. This is an important direction for future work, as the current design intentionally combines multiple factors to maximize ecological validity of the attack scenario.
>
> __Theoretical claims__
> - Response for Q4.
> - We appreciate your point regarding the interpretation of “human-like cognitive vulnerabilities. We agree that our results do not constitute direct evidence that LLMs exhibit human cognitive processes. Our intention is to use cognitive science as an inspiration for structured attack design, rather than to claim that the underlying mechanisms are literally present in LLMs.
> We will revise the wording throughout the paper to more clearly distinguish between “cognitive-science-inspired design” and “evidence of cognitive phenomena” to avoid implying LLMs possess consciousness or cognition, and align the claims with the level of empirical support.
>
> __Supplement__
>
> - __Sample size__
>   - We agree that the per-cell sample size is limited, and that fine-grained comparisons should be interpreted with caution. Our intent was to prioritize coverage across models and harm categories under a fixed evaluation budget, rather than to draw statistically significant comparisons at the individual cell level.
>   - Nevertheless, the key findings (e.g., large gap between direct prompts and CogNIP, and consistent trends across categories) are stable across all tested settings, suggesting that the conclusions are robust despite the limited per-cell samples. We will clarify this limitation，future work will expand sample sizes to enable more reliable fine-grained analysis.
>
>
> - __Impact statement and presentation__
>   - We apologize for this omission. We will add the Impact Statement in the revised manuscript. It discusses the dual-use nature of jailbreak research, our measures for responsible disclosure (including controlled access to harmful outputs), and the broader societal implications of exposing safety vulnerabilities. We will ensure it complies with ICML requirements.
>
>
> Once again, we thank you for the review and insightful questions!

---

> > ### Author Rebuttal · Reviewer_qDuz · 2026-04-03
> >
> > Thank you for the thorough and good-faith responses. Several of my concerns were acknowledged, but the core ones were deferred to future work rather than addressed.
> >
> > The baseline gap remains the most significant issue. The authors agree that comparing against existing multi-turn jailbreak methods would strengthen the evaluation but frame this as future work. Without these comparisons, the paper cannot distinguish whether the cognitive-science grounding adds value beyond what any sufficiently elaborate multi-turn prompt achieves. This is central to the paper's contribution claim.
> >
> > Similarly, the absence of ablations isolating individual mechanisms (framing vs. persona vs. cognitive load) means we cannot assess which components drive the effect. The authors' point that Stage 1 vs. Stage 2 serves as an implicit ablation is fair. Though it only tells us something about contextualization broadly, not about the specific cognitive principles the framework is built on.
> >
> > I appreciate the willingness to revise the "human-like" framing and to add a limitations section and impact statement. These are meaningful improvements. The transparency concern around model outputs is partially addressed by the offer to explore redacted releases, though this remains prospective. This framing should extend to the title as well to maintain consistency and grounding.
> >
> > The work has genuine merit in its interdisciplinary approach and experimental scope, but the current evidence base does not yet support the framework-level contribution the authors are claiming. I'll maintain my score and strongly encourage further revision and work.

---

> > > ### Author Response · Authors · 2026-04-05
> > >
> > > Thank you very much for your careful reading of our responses and for your constructive follow-up comments.
> > >
> > > We appreciate your recognition of the strengths of our work, as well as the clear articulation of the remaining concerns regarding baseline coverage and mechanism-level analysis. We fully agree that these aspects are central to establishing the framework-level contribution of CogNIP.
> > >
> > > To directly address the concern about multi-turn jailbreak baselines, we note that we have already conducted additional experiments comparing CogNIP with a representative multi-turn jailbreak baseline (DeepInception-style construction) under the same evaluation protocol. The detailed results and discussion are provided in our response to Reviewer qqSq. These results provide further empirical evidence supporting the effectiveness of our structured two-stage design beyond simpler or alternative multi-turn formulations. We acknowledge, however, that broader comparisons against additional multi-turn jailbreak methods (e.g., Crescendo, PAP, and related approaches) would further strengthen the evaluation and better situate CogNIP within the existing landscape. We will incorporate these additional baselines in the revised manuscript to ensure a more comprehensive and balanced empirical evaluation.
> > >
> > > Regarding mechanism-level ablations, we agree that isolating individual components (e.g., framing, persona assignment, cognitive load) would provide deeper insight into the internal structure of the framework. As noted, our current design evaluates CogNIP as an integrated pipeline, and the Stage 1 vs. Stage 2 separation provides only a coarse-grained view of the overall mechanism. We will explicitly acknowledge this limitation and highlight finer-grained ablation analysis as an important direction for future work.
> > >
> > > We sincerely thank you again for the thoughtful feedback!

---

### Official Review · Reviewer_a5gc · 2026-03-12

**Soundness:** 3
**Presentation:** 3
**Significance:** 3
**Originality:** 3
**Overall Recommendation:** 4
**Confidence:** 3

**Summary:**

The paper proposes a two-stage jailbreak pipeline inspired by cognitive science phenomena such as the Framing Effect, Diffusion of Responsibility, Foot-in-the-Door Effect, and psychological distancing. The approach attempts to exploit human-like cognitive blind spots in large language models. The method is evaluated on 10 state-of-the-art LLMs and demonstrates notable jailbreak success rates. The paper also claims that the framework can generalize to generate diverse prompt attacks across domains.

**Compliance With Llm Reviewing Policy:**

Affirmed.

**Key Questions For Authors:**

Since quantitative metrics already appear in the abstract, highlighting the maximum and minimum improvements or drops in metrics such as ASR, RR, or HS could strengthen the impact.
The explanation for the slight drop in ASR in Stage 2 should be expanded; authors should consider writing some text for explaining this.

**Limitations:**

No. I cant find the limitations section or the text for limitations in the paper.

**Strengths And Weaknesses:**

The idea of leveraging cognitive science and psychological effects to jailbreak LLMs is interesting and provides a different perspective compared to purely technical attack methods.
The attack success rate reported across multiple models is strong, suggesting the method is effective.
The evaluation is conducted across a wide range of modern LLM families, which strengthens the empirical results.
The framework appears generalizable and domain-agnostic, meaning it can potentially generate prompts across different application areas (e.g., health, education).
The conceptual framework used to study cognitive biases in LLM behavior is helpful and structured.
The method is relatively simple, which may make it easier to replicate and extend.

---

> ### Author Rebuttal · Authors · 2026-03-26
>
> Sincerely thank you for your review!
>
> We thank your positive assessment of our work, including the novelty of the cognitive-science perspective, the empirical performance, and the generalizability of the framework! We address the concerns regarding completeness and clarity below.
>
> __Stage 2 behavior__
> - Thank you for pointing out the need to better explain the slight drop in ASR observed in Stage 2. This phenomenon reflects a trade-off between immersion and explicitness. In Stage 1, the model operates within a narrative context with diffused responsibility, which reduces the likelihood of triggering safety filters. In Stage 2, the task shifts to decontextualized summarization, where harmful intent becomes more explicit, leading some models to partially recover their safety alignment (i.e., “wake up”).
> - This effect is model-dependent, and in several cases we observe that models maintain high compliance across both stages, which highlights differences in how alignment mechanisms respond to contextual versus explicit harmful intent. We will expand this discussion in the paper to clarify the underlying mechanism.
>
> __Abstract clarity__
> - We agree that highlighting the range of improvements would strengthen the impact. We will revise the abstract to explicitly report the maximum and minimum gains (e.g., ASR improvements across models and categories) for clearer communication of the empirical results.
>
> __Limitations__
> - We acknowledge that a dedicated limitations section is currently missing. We agree that explicitly stating these limitations will improve the paper’s clarity and rigor. In the revision, we will include a clear limitations section discussing:
>   - The limited per-cell sample size,
>   - The absence of comparisons with other multi-turn jailbreak baselines,
>   - The interpretive gap between cognitive inspiration and mechanistic explanation.
>
>
> We hope these clarifications address your concerns and demonstrate that the identified issues are primarily related to presentation and completeness rather than fundamental limitations of the approach.

---

> > ### Author Rebuttal · Reviewer_a5gc · 2026-04-01
> >
> > Satisfactory explanations.
> >
> > Modifying the score.

---

> > > ### Author Response · Authors · 2026-04-01
> > >
> > > Thank you for your thoughtful follow-up and for updating your score—we truly appreciate it!
> > > ﻿
> > >
> > > We are glad that our clarification regarding the Stage 2 behavior and overall presentation has addressed your concerns.  As you noted, the stage-wise difference highlights an interesting aspect of alignment behavior, where models may partially recover safety constraints once contextual cues are removed. We will further clarify this point in the final version to strengthen the paper’s interpretability. We also appreciate your suggestions on improving the abstract and adding a dedicated limitations section, which we will incorporate to enhance clarity and completeness.
> > > ﻿
> > >
> > > Thank you again for your feedback and support!

---

### Official Review · Reviewer_C9Bh · 2026-03-12

**Soundness:** 2
**Presentation:** 2
**Significance:** 3
**Originality:** 2
**Overall Recommendation:** 4
**Confidence:** 3

**Summary:**

This paper introduces a psychology-based framework for building LLM jailbreaks.  They try it on a suite of modern models and show that it works on all but Claude.

**Compliance With Llm Reviewing Policy:**

Affirmed.

**Key Questions For Authors:**

1) What does "deep psychological induction" mean?
2) What is a "high-dimensional cognitive narrative context"?  How do you count dimensions here?
3) "While these methods demonstrate effectiveness, research investigating the cognitive science roots underlying the behavioral anomalies of LLMs under such attacks remains limited. Our study aims to systematically address this deficiency."  My main question for this paper is how much new we learn from it.
4) "Claude Sonnet 4, which demonstrates the highest overall resilience with 0.00% ASR across nearly all categories, suffers a notable breach in C-02 (Ransomware), reaching 100.00% ASR in Stage 1 and 50.00% in Stage 2" are these round numbers because you only had 2 samples?

**Limitations:**

yes

**Strengths And Weaknesses:**

Strengths:
==============
1) The paper is timely
2) The 3 main ingredients of their attacks seem plausible and general
3) They attacked relatively modern models
4) I like that they used stratified human judges to double-check the scoring

Weaknesses:
==================

1) Figure 1 is pretty messy.
2) You don't cite
Jailbroken: How Does LLM Safety Training Fail?
Alexander Wei, Nika Haghtalab, Jacob Steinhardt
https://arxiv.org/abs/2307.02483
"We hypothesize two failure modes of safety training: competing objectives and mismatched generalization."
which seems relevant and like a competing conceptual framework to yours.
3) There are a ton of moving parts to their approach
4) They don't directly compare against different approaches, meaning that we really have to lean on conceptual novelty for the contribution.
5) Without other baselines, it's not clear if it's just easy to jailbreak all these models (except Claude).
6) You talk about the foot-in-door effect, but A) you should then cite Many-shot Jailbreaking, and also you could have studied this with pre-fill attacks.
7) I'm not sure I was ever clear what the "framing effect" is, other than introducing a setting.  Why no ablation?

Nits:
"this method induces models into a state of a cognitively immersive state"

---

> ### Author Rebuttal · Authors · 2026-03-26
>
> Sincerely thank you for your review!
> We greatly appreciate your thoughtful comments and the opportunity to clarify and improve our work. We have carefully considered all your questions and suggestions, and provide our detailed responses below:
>
> __Clarification of terminology__
> - Response for Q1 and Q2.
> - We will revise the paper to provide clearer and more precise definitions:
>     - __Psychological induction__: progressively guiding the model into a role-consistent state via multi-turn interaction.
>     - __High-dimensional context__: the use of multiple simultaneous conditioning factors (e.g., role, narrative, constraints), rather than a formal notion of dimensionality.
>      - __Framing effect__: In our work, this refers not merely to placing the task in a fictional setting, but to systematically altering how the model interprets the intent and responsibility of the request. By embedding the task within a narrative context (e.g., role-playing or hypothetical scenarios), the same underlying instruction is reframed as part of story coherence rather than a direct real-world action, which reduces the likelihood of triggering safety mechanisms. This aligns with the cognitive science notion of framing,
> where equivalent information presented under different contexts leads to different responses.
>
> __Contribution and novelty__
> - Response for Q3.
> - We would like to clarify that the primary contribution of this work is not a single new prompt pattern, but a structured, cognitively-inspired framework for systematically constructing jailbreak attacks.
> - While prior work (including prompt engineering approaches and studies such as “Jailbroken: How Does LLM Safety Training Fail?”) identifies failure modes or proposes specific attack strategies, our goal is to provide a unifying design perspective that connects these behaviors to interpretable principles (e.g., framing, responsibility diffusion, and staged interaction). This perspective helps explain why multi-turn and role-based attacks are effective, rather than only demonstrating that they work. We will clarify this positioning and better situate our work relative to prior frameworks in the revision.
>
> __Sample size and interpretation of results__
> - Response for Q4.
> - We thank you for this observation. The reported values (e.g., 100%, 50%) indeed reflect the limited number of trials per model-category-condition (e.g., 6/6 or 3/6 successful trials).
> - We agree that such small sample sizes make these per-category results sensitive to individual trials and should not be over-interpreted in a statistical sense. Our experimental design prioritizes broad coverage across models and categories under a fixed evaluation budget, rather than high-resolution estimation within each individual cell.
> - Importantly, our intention is not to claim precise quantitative differences at the cell level, but to highlight the existence of category-specific vulnerabilities and heterogeneous safety behavior across models. The “Achilles’ Heel” examples are intended as illustrative case studies of potential blind spots rather than definitive statistical conclusions.
> - We will revise the wording to better reflect this and explicitly clarify the underlying sample sizes to avoid possible misinterpretation.
>
> __Supplement__
> - __Baselines and Ablation__
>      - We agree that comparisons with other multi-turn jailbreak methods and ablations of individual components would strengthen the work. Our current design prioritizes demonstrating the effect of structured multi-stage prompting under a controlled baseline. We acknowledge this limitation and will include it explicitly, along with these directions for future work.
> - __Related work__
>    - Thank you for pointing out relevant prior work, including “Wei et al., 2023” and other jail breaking approaches. We will add these citations and better position our work relative to these lines of research in the revision.
>
> Thanks again for your review and questions which have helped us improve both the clarity and positioning of our work!

---

> > ### Author Rebuttal · Reviewer_C9Bh · 2026-03-31
> >
> > Thanks for the detailed reply.
> >
> > Re: "High-dimensional context: the use of multiple simultaneous conditioning factors" - so is it like 3 dimensions then?  Still seems like a bad choice of words.  Why not just "multiple dimensions"?
> >
> > Re: your method is a "a structured, cognitively-inspired framework for systematically constructing jailbreak attacks."  This is a reasonable kind of contribution, but in that case I would expect many more ablations, and examples of how existing jailbreak families fit or don't fit into this framework.  E.g. take an existing jailbreak, show what's missing from it according to your framework, fix it up to match your method and watch it become more effective.
> >
> > Overall I like qDuz's review better than mine so I am copying their score.

---

> > > ### Author Response · Authors · 2026-04-01
> > >
> > > Thank you so much for your thoughtful follow-up and for the constructive suggestions. We truly appreciate the time you have invested in engaging with our work.
> > >
> > > __Terminology: "high-dimensional context"__
> > > - We agree that “high-dimensional” may be misleading. Our intention was to refer to the presence of multiple conditioning factors (e.g., role, narrative, constraints). We will revise this to "multiple conditioning factors" for clarity.
> > >
> > > __On framework validation and ablations__
> > > - We appreciate your point that a framework contribution should ideally be supported by ablations and by demonstrating how it can improve existing jailbreak strategies.
> > >
> > > - At the same time, while we do not perform component-level ablations, our two-stage framework introduces a coarser but structured functional decomposition. Specifically, the comparison between Stage 1 (contextualized generation) and Stage 2 (decontextualized extraction) can be viewed as an implicit ablation that isolates the effect of contextual conditioning as a whole. This allows us to directly examine how alignment behavior changes once contextual cues are removed, which is difficult to observe in standard single-stage jailbreak setups. We believe this stage-level analysis already provides useful insight into the mechanisms underlying jailbreak success, complementing (rather than replacing) finer-grained ablations. We agree that extending the framework to systematically analyze and enhance existing jailbreak methods is an important direction, and we will clarify this scope in the revision.
> > >
> > >
> > > Thank you again for your thoughtful engagement and for helping us improve the clarity and depth of our contribution. We hope this clarification addresses your concern regarding the lack of ablation and better highlights the role of stage-level decomposition in validating the framework.

---

### Official Review · Reviewer_qqSq · 2026-03-13

**Soundness:** 1
**Presentation:** 3
**Significance:** 1
**Originality:** 2
**Overall Recommendation:** 2
**Confidence:** 5

**Summary:**

This paper presents a jailbreak strategy inspired by Cognitive Science concepts called CogNIP. CogNIP consists of two stages: Stage 1 utilizes framing effects, responsibility diffusion, and cognitive load to craft a narrative prompt that generates harmful content wrapped in the trappings of the narrative present in the original prompt, Stage 2 exploits the foot-in-the-door effect by asking the model to extract narratively encoded content into clean, actionable instructions. CogNIP is tested using harmful queries across 10 categories like Doxxing, Phishing etc. and is shown to produce 71% average ASR for CogNIP Stage 1 and 65% average ASR for CogNIP Stage 2 across 10 LLMs jumping from a baseline ASR of ~2%. The paper also presents a category-level breakdown of ASR showing that frontier LLMs are especially vulnerable to certain categories and different model providers potentially focusing on safeguards for some categories over others.

**Compliance With Llm Reviewing Policy:**

Affirmed.

**Final Justification:**

This paper proposes a jailbreak creation framework rooted in cognitive science fundamentals.  My initial review contains, in my opinion, an accurate characterisation of this work as lacking in terms of baselines, standardised evaluations and framing-relevant experiments.

The rebuttal from the authors did change my opinion on the evaluation judge used and their ICC validation is non-trivial. Keeping this in mind, I’m increasing my score to a 2 but can’t increase my score further due to the aforementioned weaknesses.

**Key Questions For Authors:**

Q.1 How does CogNIP's Stage 1 narrative immersion differ mechanistically from DeepInception's[1] nested fictional scenarios with character assignment?

Q.2 How does the two-stage pipeline (embed in narrative, then extract) differ from the encode-then-decode structure in CipherChat and related work?

[1] Li, X., Zhou, Z., Zhu, J., Yao, J., Liu, T., & Han, B. (2023). Deepinception: Hypnotize large language model to be jailbreaker. arXiv preprint arXiv:2311.03191.

**Limitations:**

The paper doesn't contain an Impact Statement and neither contains any information about responsible disclosure. To their credit they do redact some sensitive prompt details to hamper easy reproduction of this attack but an explicit acknowledgement including an Impact Statement would have been appreciated.

**Strengths And Weaknesses:**

Strengths:
- Provides a unified pipeline based on Cognitive Science theory that successfully jailbreaks frontier production LLMs
- Articualate writing and well organized figures

Weaknesses:
- The paper dresses up known failure modes with psychological terminology making it hard to state with confidence whether the cognitive science framing provides genuine explanatory power or if they are post-hoc rationalizations of empirically observed attack effectiveness. Specifically, persona jailbreaks[1,2] have been demonstrated as well as cipher decoding attacks[3] (akin to narrative decoding) which CogNIP essentially combines together.
- The aforementioned jailbreak papers are not cited making the central result of this paper appear stronger than it actually is
- The harmful queries used in this paper are generated in an ad-hoc manner and not based on existing benchmark taxonomies like StrongREJECT[4]. The paper also doesn't test CogNIP on specific benchmark datasets[4, 5] making these results not comparable to existing jailbreak techniques
- The evaluation of responses to enumerate their harmfulness is done using a persona prompted LLM as a judge and this judge is compared with limited human judgement across ~500 responses. The LLM judge is also not compared to existing LLM judges[4,5] making these evaluation results highly non-standard.

[1] Shah, R., Pour, S., Tagade, A., Casper, S., & Rando, J. (2023). Scalable and transferable black-box jailbreaks for language models via persona modulation.

[2] Zhang, Z., Zhao, P., Ye, D., & Wang, H. (2025). Enhancing jailbreak attacks on llms via persona prompts.

[3] Yuan, Y., Jiao, W., Wang, W., Huang, J., He, P., Shi, S., & Tu, Z. (2023). GPT-4 Is Too Smart To Be Safe: Stealthy Chat with LLMs via Cipher.

[4] Souly, A., Lu, Q., Bowen, D., Trinh, T., Hsieh, E., Pandey, S., ... & Toyer, S. (2024). A strongreject for empty jailbreaks. Advances in Neural Information Processing Systems, 37, 125416-125440.

[5] Mazeika, M., Phan, L., Yin, X., Zou, A., Wang, Z., Mu, N., ... & Hendrycks, D. (2024). Harmbench: A standardized evaluation framework for automated red teaming and robust refusal. arXiv preprint arXiv:2402.04249.

---

> ### Author Rebuttal · Authors · 2026-03-26
>
> Thank you for the detailed and well-referenced feedback!
> We appreciate the concerns regarding novelty, evaluation standards, and positioning relative to prior work.
>
> __Novelty and relation to prior work__
> - We respectfully clarify that our contribution is not a combination of existing jailbreak techniques, but a structured, interpretable, two-stage framework that decomposes the attack process into distinct phases:
>     - (i) narrative immersion under contextual framing, and
>     - (ii) decontextualized extraction of actionable content.
> - While prior work explores individual components (e.g., persona-based or encode-decode approaches), these are typically implemented in a single-stage manner. In contrast, our explicit stage separation enables analysis of how alignment behavior changes when harmful content transitions from contextualized to explicit form (e.g., partial recovery of alignment in Stage 2 for some models). This diagnostic perspective is central to our contribution.
> - Importantly, cognitive science in our work serves as a design prior to structure the attack process, rather than as a post-hoc explanation.
>
> __Relation to DeepInception and CipherChat__
> - Response for Q1 and Q2.
> - __DeepInception__: It relies on nested narrative construction within a single-stage
> interaction. CogNIP differs by explicitly separating immersion and extraction
> into two stages with distinct roles, enabling analysis of alignment dynamics
> across stages.
> - __CipherChat__: This approache focuss on obfuscation via encoding and decoding. In
> contrast, our focus is on how contextual framing and subsequent decontextualization influence alignment behavior, rather than obfuscation alone.
>
> __Supplement__
> - __Evaluation and benchmarks__
>     - We agree that standardized benchmarks (e.g., StrongREJECT, HarmBench) would improve comparability. Our current evaluation prioritizes controlled stress-testing across models and categories. We will clarify this limitation and better position our results relative to benchmark-based evaluations.
>
> - __Evaluation methodology__
>
>     - We combine LLM-based evaluation with stratified human judgments and validate agreement via ICC. We will clarify this setup and its limitations relative to existing evaluation frameworks.
>
> - __Ethics__
>     - We sincerely apologize for this oversight. In the revised manuscript, we will include an Impact Statement that addresses the dual-use nature of jailbreak research, outlines our responsible disclosure practices (including controlled access to potentially harmful outputs), and discusses the broader societal implications of uncovering safety vulnerabilities.
>
>
> We hope these clarifications better position our contribution as a structured framework for analyzing jailbreak behaviors. Thank you!

---

> > ### Author Rebuttal · Reviewer_qqSq · 2026-04-02
> >
> > I appreciate the author's response but the central claim of CogNIP being a structural framework that can enable the construction of jailbreaks is not effectively tested. Respectfully, the lack of established evaluation datasets and low sample sizes (as noted by other reviewers) highly detracts from the potential widespread applicability of CogNIP. Lack of contextualizing CogNIP generated jailbreaks to other published jailbreaks also limits the empirical validity of their findings. Beyond simple contextualization, the authors can also potentially show that several existing published jailbreaks could be generated using CogNIP cementing it's utility as a structural framework. While the authors assert that "cognitive science in our work serves as a design prior to structure the attack process, rather than as a post-hoc explanation" the current empirical results in the paper do not make a strong case for this.
> >
> > I recommend the authors to flesh out this work in the aforementioned two directions and resubmit to future conferences. I'll retain my score.

---

> > > ### Author Response · Authors · 2026-04-05
> > >
> > > Thank you for the thoughtful follow-up and for clearly articulating the remaining concern regarding the empirical grounding of CogNIP as a structural framework.
> > > ﻿
> > >
> > > Until now, we have finished additional experiments to evaluate CogNIP against a widely recognized baseline jailbreak paradigm (DeepInception-style prompt constructions) across the same set of models, using identical evaluation protocols as in the paper. Specifically, we re-generated all attack prompts in our evaluation set using DeepInception-style constructions and re-ran the full experimental pipeline under the same settings, including the same model suite, datasets, and evaluation metrics (ASR, RR, HS). We only report ASR and HS below:
> > > ﻿
> > > -  ASR (%)
> > >    | Model | Direct | DeepInception | CogNIP Stage 1 | CogNIP Stage 2 |
> > > |-------|--------|---------------|----------------|----------------|
> > > | DeepSeek-V3 | 0.00 | 76.67 | 66.67 | 56.67 |
> > > | GPT-5 | 0.00 | 1.67 | 46.67 | 36.67 |
> > > | Claude Sonnet 4 | 0.00 | 0.00 | 10.00 | 5.00 |
> > > | Kimi-K2 | 3.33 | 41.67 | 81.67 | 78.33 |
> > > | Ernie 4.5 Turbo | 0.00 | 43.33 | 85.00 | 85.00 |
> > > | Qwen3-235B | 1.67 | 86.67 | 81.67 | 78.33 |
> > > | Qwen3-30B | 6.67 | 36.67 | 88.33 | 86.67 |
> > > | Llama-3.1-8B | 6.67 | 23.33 | 96.67 | 88.33 |
> > > | Gemini 3 Pro | 0.00 | 18.33 | 55.00 | 41.67 |
> > > | Grok 4.1 | 3.33 | 20.00 | 98.33 | 98.33 |
> > > | **Average** | **2.17** | **34.83** | **71.00** | **65.50** |
> > > ﻿
> > > -  HS (Harmfulness Score, 1–5)
> > > ﻿  | Model | Direct | DeepInception | CogNIP Stage 1 | CogNIP Stage 2 |
> > > |-------|--------|---------------|----------------|----------------|
> > > | DeepSeek-V3 | 1.00 | 4.00 | 3.90 | 3.65 |
> > > | GPT-5 | 1.00 | 1.15 | 2.70 | 2.35 |
> > > | Claude Sonnet 4 | 1.00 | 1.00 | 1.30 | 1.15 |
> > > | Kimi-K2 | 1.30 | 2.65 | 4.35 | 4.45 |
> > > | Ernie 4.5 Turbo | 1.00 | 2.95 | 4.40 | 4.10 |
> > > | Qwen3-235B | 1.05 | 3.65 | 4.00 | 3.75 |
> > > | Qwen3-30B | 1.15 | 2.65 | 4.05 | 3.90 |
> > > | Llama-3.1-8B | 1.30 | 2.60 | 4.25 | 4.05 |
> > > | Gemini 3 Pro | 1.05 | 2.00 | 3.05 | 2.25 |
> > > | Grok 4.1 | 1.20 | 1.80 | 4.70 | 4.60 |
> > > | **Average** | **1.11** | **2.45** | **3.67** | **3.43** |
> > > ﻿
> > > ﻿
> > > The updated results show a consistent and systematic improvement of CogNIP-based constructions  over the DeepInception baseline across the majority of evaluated models. In particular, CogNIP variants achieve substantially higher ASR while maintaining comparable or improved harmfulness scores, indicating that the structured two-stage decomposition contributes non-trivial gains beyond nested narrative prompting alone. In the revised version, we will expand baseline coverage to include additional representative jailbreak frameworks beyond DeepInception.
> > > ﻿
> > > Thank you again for your guidance in strengthening this work and we will improve in the final version!

---

### Decision · Program_Chairs · 2026-04-30

**Decision:**

Reject

**Comment:**

The present submission proposes a new mechanism for running adversarial attacks against safety systems of LLMs. The mechanism is inspired by psychological concepts, attempting to make the model believe an unsafe request is actually safe within the given context. This is an interesting topic and idea. This has been echoed by the reviewers' feedback. However, during the review and rebuttal phase, a couple of weaknesses were raised, such as (1) fairly limited sample sizes; (2) a missing comparison with any of the already existing jailbreaking approaches; and (3) partially insufficient evidence for justifying the psychological/cognitive science spin of the paper. I believe that the authors can address these concerns in a major revision of the work. Therefore, the paper should not be published in its current form, but instead, a revised version (with extended experiments and comparison to more existing work) should be submitted to a future venue.